# Necrotic cardiac myocytes skew macrophage polarization towards a classically activated phenotype

Wenlong Jiang[1,2]*, Luigi Adamo[3], Kenji Lim[1], Scot J Matkovich[1], Sarah Evans[1], Cibele Rocha-Resende[1], Douglas L Mann[1]

1 Center for Cardiovascular Research, Washington University School of Medicine, St. Louis, MO, United States of America, 2 Cardiology, Jiangyin's People Hospital, Jiangyin, Jiangsu, China, 3 Division of Cardiology, Johns Hopkins University School of Medicine, Baltimore, Maryland, United States of America

* jiangwlcardio@163.com

**Data Availability Statement:** All relevant data are within the manuscript and its Supporting Information files.

## Abstract

Necrotic and dying cells release damage-associated molecular patterns (DAMPs) that can initiate sterile inflammatory responses in the heart. Although macrophages are essential for myocardial repair and regeneration, the effect of DAMPs on macrophage activation remains unclear. To address this gap in knowledge we studied the effect of necrotic cardiac myocyte extracts on primary peritoneal macrophage (PPM) cultures in vitro. We first performed unbiased transcriptomic profiling with RNA-sequencing of PPMs cultured for up to 72 hours in the presence and absence of: 1) necrotic cell extracts (NCEs) from necrotic cardiac myocytes in order to mimic the release of DAMPs; 2) lipopolysaccharide (LPS), which is known to polarize macrophages towards a classically activated phenotype and 3) Interleukin-4 (IL-4), which is known to promote polarization of macrophages towards an alternatively activated phenotype. NCEs provoke changes in differential gene expression (DEGs) that had considerable overlap with LPS-induced changes, suggesting that NCEs promote macrophage polarization towards a classically activated phenotype. Treating NCEs with proteinase-K abolished the effects of NCEs on macrophage activation, whereas NCE treatment with DNase and RNase did not affect macrophage activation. Stimulation of macrophage cultures with NCEs and LPS resulted in a significant increase in macrophage phagocytosis and interleukin-1β secretion, whereas treatment with IL-4 had no significant effect on phagocytosis and interleukin-1β. Taken together, our findings suggest that proteins released from necrotic cardiac myocytes are sufficient to skew the polarization of macrophages towards a classically activated phenotype.

## Introduction

Acute Myocardial Infarction (AMI) and other forms of sterile tissue injury initiate a series of complex myocardial inflammatory responses that can be divided into two overlapping phases: an acute inflammatory phase and a subsequent reparative phase [1–4]. The inflammatory

**Funding:** Dr. Adamo was supported by NIH grant 1K08HL145108-01. But the funders had no role in study design, data collection and analysis, decision to publish, or preparation of the manuscript.

**Competing interests:** The authors have declared that no competing interests exist.

phase is characterized by activation of the innate immune response and infiltration of bone marrow -derived circulating leukocytes. This initial inflammatory phase is thought to promote the clearing of matrix debris and dead cells from the infarct site. The second, reparative phase of the inflammatory response is characterized by the differentiation/polarization of leukocytes and other cells to ramp up functions associated with resolution of the inflammatory response and the initiation of scar formation [5].

Macrophages and monocytes play a key role in both the inflammatory phase and the reparative phase of tissue injury [6–8]. Following tissue injury, monocyte derived macrophages that enter the heart manifest a classically activated (M1) phenotype, that is characterized by expression of pro-inflammatory mediators such as nitric oxide synthase 2 (NOS2), tumor necrosis factor (TNF), interleukin-1β (L-1β), and IL-6 [7, 8]. During the reparative phase of tissue injury, macrophages acquire a different phenotype that is characterized by the expression of anti-inflammatory mediators such as transforming growth factor β (TGF-β) and IL-10. This macrophage phenotype has been referred to as the alternatively activated (M2) phenotype [9, 10]. The molecular cues that skew the polarization of macrophages towards a classically activated or an alternatively activated phenotype in the setting of acute myocardial injury are not completely understood.

Following tissue injury, necrotic and dying cardiac myocytes release their cytosolic contents into the extracellular space, which in turn initiates a brisk inflammatory response. The proteins that are released by dead or dying cells have been referred to as damage associated molecular patterns (DAMPs) [11, 12]. Mammalian cells express evolutionarily conserved pattern recognition receptors (PRRs) that are capable of recognizing conserved molecular motifs contained in DAMPs [11]. When DAMPs engage PRRs they elicit inflammatory responses that are essential for proper initiation and resolution of tissue injury [13]. Macrophages are known to play an essential role in myocardial tissue injury and repair [14–16]. However, the role that DAMPs play in regulating macrophage biology is not well understood. Here we show that proteins in necrotic cell extracts are sufficient to skew the polarization of macrophages towards a classically activated phenotype.

## Methods

### Study approval

All experimental procedures were performed in accordance with approved animal protocols from the Institutional Animal Care and Use Committee at Washington University School of Medicine. These investigations conform to the NIH Guide for the Care and Use of Laboratory Animals.

### Peritoneal macrophage culture

The isolation of primary intraperitoneal macrophages was performed as previously described In brief, C57BL/6 mice (8–10 weeks) were injected intraperitoneally with 1 ml thioglycolate broth to elicit intraperitoneal accumulation of macrophages. Mice were euthanized 4 days later and inflammatory cells were harvested from the peritoneal cavity by washing with 10 ml of cell culture media. Peritoneal-derived cells were plated onto 24-well plates and cultured in high-glucose DMEM containing 10% FBS, 2 mM L-glutamine, 100 U/ml penicillin, and 0.1 mg/ml streptomycin. For the gene expression studies, cells were plated at a density of 250,000 cells per well. For macrophage phagocytosis assay cells were plated at a density of 125,000 per well. After 12–24 hours of plating non-adherent cells were removed washing twice with PBS and fresh cell culture medium was added. Culture medium was replaced every 3 days, as needed.

## Preparation of necrotic cardiac myocyte cell extracts

Necrotic cardiac myocyte cell extracts (NCEs) were obtained using a modification of the method by Beyer and Pisetsky [12]. H9C2 embryonic cardiac myocyte cells were cultured in T75 tissue culture flasks in DMEM high-glucose medium supplemented with 10% fetal bovine serum, 100 U/mL penicillin, 100 μg/mL streptomycin. Culture media was replaced every 3 days. Cells were passaged at 80–90% confluence for a maximum of 10 passages. H9C2 cells were harvested, pelleted and necrosis was induced by freeze-thawing for 5 cycles (10 minutes in -80˚C freezer followed by 3 minutes in a water bath at 37˚C), followed by centrifugation at 12 000g for 10 minutes at 4˚C. The supernatant was collected and stored at -80˚C in aliquots. Each batch of H9C2 cells used to generate cytosolic extracts were tested to rule out myco-plasma contamination and cytosolic extracts were confirmed to be endotoxin-free by LAL Chromogenic Endotoxin Quantitation Kit (Thermo Fisher Scientific, Catalog No:88282).

To determine whether the biological effects of NCEs were secondary to proteins, RNA, or DNA, the NCEs were digested with proteinase K (PK) RNase and DNase. NCEs and controls (plasmid DNA [1 μg] and total RNA [1 μg]) were treated with DNase I (5 μg/mL) or RNase (5 μg/mL) and incubated in a water bath at 37˚ C for 60 min. Effective digestion of nucleic acids was verified by agar gel electrophoresis of plasmid DNA and RNA. NCEs were treated with PK (100 μg/ml) and incubated in a water bath at 55˚C for 60 min before heating to 95˚C for 10 minutes. Effective digestion of protein was confirmed using the BCA assay.

## Transcriptional profiling of macrophage cultures

To begin to understand how NCEs influence macrophage biology, we examined time-dependent changes in gene transcription in macrophage cell cultures that were stimulated with diluent (DMEM), 10 μg/ml NCEs, 100 ng/ml of lipopolysaccharide (LPS), or 10 ng/ml interleukin-4 (IL-4) for 8, 24 and 72 hours (n = 4 cultures/condition). At the end of the treatment period, the macrophage cultures were washed twice with cold PBS and then lysed in RNA lysis buffer for RNA extraction and gene expression analysis. For each condition, the RNA from 4 separate macrophage cultures was collected and pooled. Total RNA was extracted using the ZYMO research Quick-RNA MicroPrep kit according to the manufacturer's instructions (Catalog No: R1051). Total RNA was selected for polyadenylated RNA and converted to RNA-sequencing libraries using the SMARTer v2 kit from Clontech. Single-end, 50-bp reads were obtained on an Illumina HiSeq 3000 and aligned to the Illumina iGenomes GRCm38_Ensembl release of the mouse transcriptome using Tophat2.1 [17] yielding an average of $1.8 \times 10^7$ aligned reads per sample. Gene-level quantification was performed using HTSeq 2 [18]. The limma-voom procedure was used to identify mRNAs with differential expression between experimental conditions (#22303). Differentially expressed genes between diluent and

NCE, LPS and IL-4 treated cultures were determined using a step-up false discovery rate (FDR) < 0.01 and an absolute fold change of ≥ 2.

*Real-time quantitative PCR* Total RNA was extracted using the ZYMO research Quick-RNA MicroPrep kit according to the manufacturer's instructions (Catalog No: R1051). RNA concentration was determined by Nanodrop. cDNA was synthesized using BIO-RAD iScript™ Reverse Transcription Supermix or Applied Biosystems™ High-Capacity RNA-to-cDNA™ Kit. Quantitative PCR for IL-6 (*Il6*), TNF (*Tnf*), NOS2 (*Nos2*), macrophage galactose-type lectin (*Mgl*), Chi3l3 (*Chi3l3*), and resistin-like alpha (*Retnla*) was performed using the PowerUp™ SYBR™ Green Master Mix (Applied Biosystems™) on a QuantStudio™ 3 Real-Time PCR System (Applied Biosystems™). TaqMan Gene Expression assays (Thermo Fisher) were used for qPCR for tissue growth factor-β1(*Tgfb* [Mm01178820_m1] and Interleukin-10 (*Il10* [Mm01288386_m1]). Primer sequences for each gene studied are listed in **Table 1**. Gene

**Table 1. List of primers for PCR.**

| Gene | Primer | Sequence (5'-3') | Length (bases) |
|---|---|---|---|
| *Il6* | Forward | TCCTTAGCCACTCCTTCTGT | 20 |
| | Reverse | AGCCAGAGTCCTTCAGAGA | 19 |
| Tnf | Forward | GGTGCCTATGTCTCAGCCTCTT | 22 |
| | Reverse | GCCATAGAACTGATGAGAGGGAG | 23 |
| Nos2 | Forward | CACTTCTGCTCCAAATCCAAC | 21 |
| | Reverse | GACTGAGCTGTTAGAGACACTT | 22 |
| Mgl | Forward | GACCAAGGAGAGTGCTAGAAG | 21 |
| | Reverse | TGACTGAGTTCCTGCCTCT | 19 |
| Chi3l3 | Forward | ACTGGTATAGTAGCACATCAGC | 19 |
| | Reverse | AGAAGCAATCCTGAAGACACC | 21 |
| Retnla | Forward | CACACCCAGTAGCAGTCATC | 20 |
| | Reverse | TGCCAATCCAGCTAACTATCC | 21 |
| 36B4 | Forward | AGGGCGACCTGGAAGTCC | 18 |
| | Reverse | CCCACAATGAAGCATTTTG GA | 21 |

Il6, Interleukin 6; Tnf, Tumor Necrosis Factor; Nos2, Nitric Oxide Synthase 2; Mgl, macrophage galactose-type lectin; Chi3l3, Chitinase 3-like-3; Retnla, Resistin-like molecule alpha; 36B4, the housekeeping gene.

expression values were calculated with the comparative 2-ΔΔCT method using expression levels of 36B4 *mRNA* as an endogenous control.

## Measurement of interleukin-1β

Macrophage culture media was collected at 8, 24, or 72 after treatment with diluent, NCEs, LPS, or IL-4, centrifuged at 3000 g for 5 minutes to remove cell debris, and the supernatants were subsequently aliquoted and stored at -80˚C. IL-1β levels in the culture medium supernatants were measured by ELISA using commercially available kits (Sigma-Aldrich, MO, USA, # EM2IL1B), according to the manufacturer's instructions. All the samples were thawed only once and all standards and samples were run in duplicates.

## Phagocytosis assay

The phagocytic capacity of macrophages was assessed by measuring particle uptake as described previously [19]. Briefly, 0.5um Fluoresbrite® Yellow-green fluorescent polystyrene microspheres (Polysciences Inc., Warrington, PA) were suspended in PBS, vortexed briefly, and centrifuged at 10,000g for 8 min x2, before incubating them for 30 minutes at 37 ˚C in 50% FBS. After this preparatory step, beads were added to pre-stimulated macrophage cultures were plated on coverslips in 24 well plates at a density of 100,000 cells/well. The bead/cell ratio was optimized to be approximately 200:1. After 2 hours of incubation, coverslips were rinsed with ice-cold PBS x3, fixed with 4% paraformaldehyde, and covered with aluminum foil to protect the fluorescent beads from bleaching. Before imaging cells were immunostained with anti-CD68 as the primary antibody (Bio-Rad. Cat: MCA1957) and Goat anti-rat 594 as the secondary antibody (InvivoGen, Cat: A11007). DAPI was used to stain the nuclei. Images were acquired using a Confocal Scanning Microscope (LSM 800) with the appropriate filters. The phagocytic activity was calculated by counting the CD68+ cells uptaking of the fluorescent beads in 10 fields and then divided the total number of cells present in the same field.

### Reactive oxygen species assay

To evaluate the amount of reactive oxygen species produced in primary peritoneal macrophages, a 2',7'–dichlorofluorescin diacetate (DCFDA) assay was performed (Abcam, ab113851) following manufacturer's instructions. Briefly, 50,000 peritoneum-derived cells were seeded on black, flat clear bottom 96-well plates (Corning, 3603). The following day, non-adherent cells were washed off and peritoneal macrophages were given the same culture medium as previously described but without phenol red, and then either unstimulated or stimulated with 10 μg/mL of mouse-heart derived NCE. After 24 hr, DCFDA was prepared in the medium used for NCE stimulation, added to cells at a final concentration of 20 μM, and incubated at 37˚C in the dark for 45–60 min. Fluorescence levels were immediately measured using a Tecan Infinite M200 Pro microplate reader at excitation and emission wavelengths of 485 nm and 535 nm, respectively. Plates were read from the bottom; multiple reads were done per well using the circle (4x4) filled option with 1400 μm border.

### Statistical analysis

Data are presented as average ± standard error of the mean (SEM). Statistical calculations were performed using GraphPad Prism version 8 (GraphPad Software Inc., La Jolla, CA, USA). To evaluate differences in cytokine concentration we used ANOVA followed by Dunnett's multiple comparisons test. Chi-squared test was used to compare phagocytosis ability of macrophages across different conditions. A value of $P< 0.05$ was considered significant.

## Results

### Necrotic cell extracts skew macrophage polarization towards a classically activated phenotype

To explore the effects of NCEs on macrophage biology we performed unbiased transcriptional profiling of primary peritoneal macrophage cultures that were treated with NCEs in vitro for 8, 24, 72 hours. The NCE-induced changes in gene expression were compared to changes in gene expression in macrophage cultures stimulated for identical times with LPS, a known stimulus for classic activation of macrophages, or with IL-4, a known stimulus for activation of alternatively activated macrophages. Diluent treated macrophage cultures served as the appropriate control group for the NCE, LPS, and IL-4 stimulated cultures.

As shown in **Fig 1A**, after 8 hours of stimulation the number of unique differentially expressed genes (fold-change $\geq$ 2, FDR <0.01) was greatest for LPS (1878 DEGs) > NCE (339 DEGs) > IL-4 (138 DEGs). Similar trends were also observed at 24 hours: LPS (1146 DEGs) > NCE (194 DEGs) > IL-4 (138 DEGs). **Fig 1B** shows that there was considerable overlap of gene expression between NCE and LPS at 8 and 24 hours, whereas there was minimal overlap with NCE and IL-4, suggesting that NCEs skew the macrophage phenotype towards a classical pattern of activation. After 72 hours of stimulation, the number of DEGs was greatest for LPS (1010 genes) > IL-4 (384 genes) > NCE's (192 genes). Intriguingly, after 72 hours the number of overlapping genes was ~ 50% less for NCEs and LPS and was substantially increased for NCEs, LPS, and IL-4 (**Fig 1B**) suggesting that NCE-induced macrophage polarization towards a classical pattern diminishes over time and that after 72 hours of stimulation macrophage polarization is more towards an alternatively activated pathway for both LPS and NCE. A gene ontology analysis of the top 10 pathways for genes that were differentially expressed following acute (8 hour) stimulation with NCEs or LPS revealed pathways that were enriched for biological themes reflecting the cellular response to stress and inflammation (**Fig 1C**), which are consistent with macrophage polarization towards a classically activated phenotype (see **S1 File** for

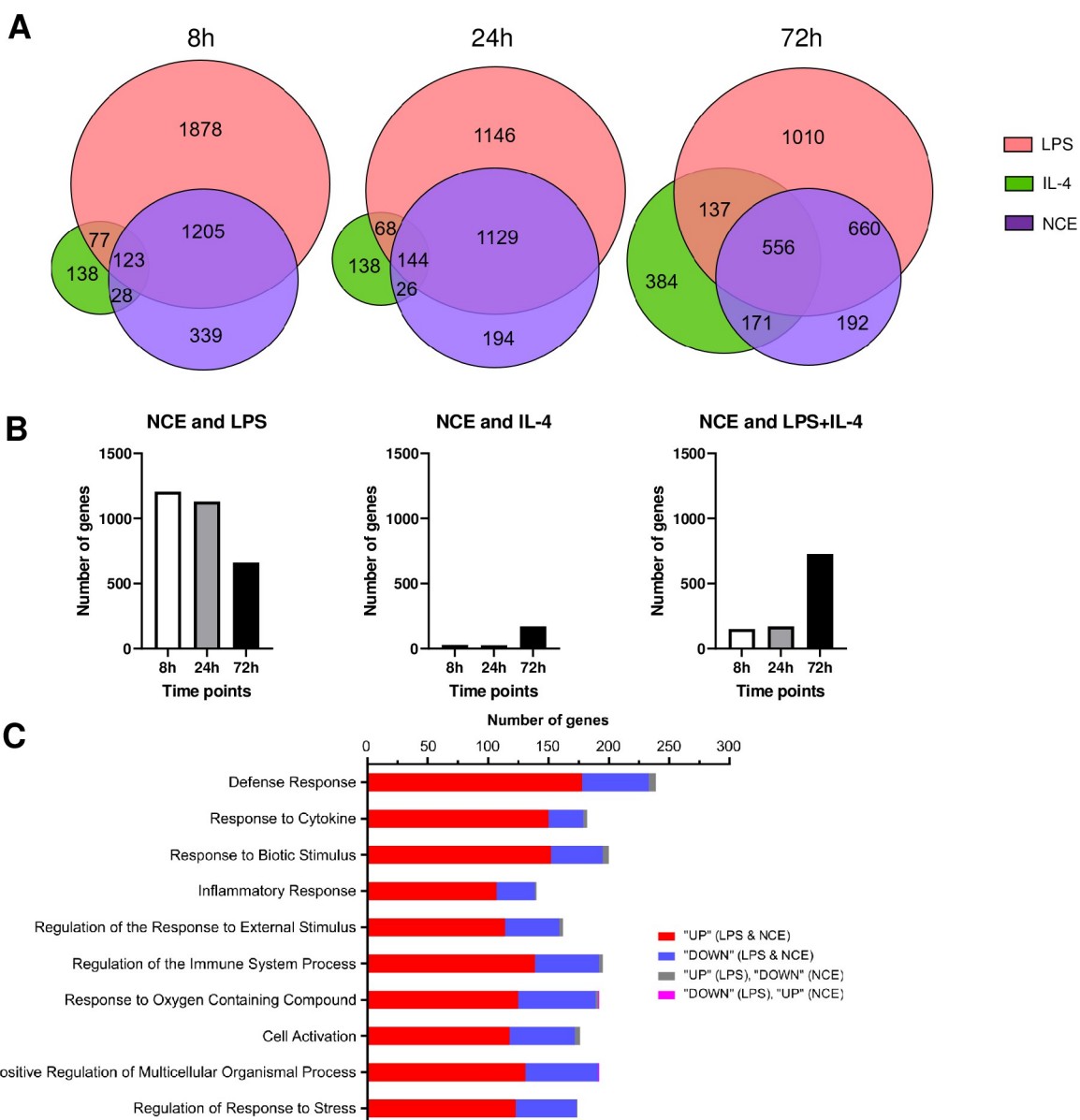

**Fig 1. Unbiased transcriptional profiling of primary peritoneal macrophage cultures.** (A) Primary peritoneal macrophages were exposed to necrotic cell extracts (NCEs), lipopolysaccharide (LPS) or interleukin-4 (IL-4) for 8, 24 or 72 hours. For each condition and time point, differentially expressed genes (DEGs) were determined in comparison to diluent treated cell cultures. The Venn diagrams show the overlap of differentially expressed gene between the 3 treatments at 8h, 24h, 72h. (B) the number of DEGs that were common to NCEs and LPS, NCEs and IL-4 and NCEs and LPS + IL-4 are show at 8, 24 or 72 hours. (C) Gene ontogeny (GO) analysis genes that were acutely changed in NCE and LPS treated cultures.

a list of differentially expressed genes that were up and down regulated following stimulation with NCEs or LPS).

To validate the findings from the RNAseq analysis we performed quantitative real-time PCR (qPCR) of selected genes in the LPS, NCE and IL-4 stimulated macrophage cultures. Based on the previous reports, We determined mRNA levels for *Il6*, *Tnf*, and *Nos2* as genes that are associated with the classically activated macrophages, and determined mRNA levels for *Retnla, Chi3l3, Mgl, Tgfb1 and Il10* as genes associated with alternatively activation of

macrophages. As shown in **Fig 2**, NCE treatment significantly (p < 0.001) increased mRNA levels for *Il6*, *Tnf*, and *Nos2*, and significantly (p < 0.05) decreased mRNA levels for *Mgl* and *Retnla*, but had no effect on *Chi3l3*, *Tgfb1 and Il10* mRNA levels. These findings are consistent overall with the transcriptional profiling studies and indicate that NCEs skew macrophage polarization towards a more classically activated phenotype.

To determine whether the effects of NCEs on macrophage polarization were mediated by proteins, DNA, or RNA contained in the cardiac myocyte extracts, the NCEs were pre-digested with proteinase K, DNase I, or RNase prior to treating the macrophage cultures for 24 hours. As shown in **Fig 3A**, digestion of NCEs with proteinase K completely abolished NCE-induced upregulation of IL-6 mRNA, which was used as a marker of classical activation of macrophages. In contrast treatment with DNase (**Fig 3B**) and RNase (**Fig 3C**) had no effect on NCE-induced upregulation of IL-6 mRNA, suggesting that proteins within the NCEs are responsible for macrophage polarization.

## Functional significance of necrotic cell extract macrophage polarization

To understand the functional significance of the changes in NCE-induced gene expression changes, we examined phagocytosis and IL-1β secretion in macrophage cultures that were stimulated with LPS, NCEs, or IL-4. To assess phagocytosis, we pre-treated macrophage cultures with diluent (control medium), LPS, NCEs, and IL-4 for 24 hours and then incubated the cultures with fluorescent beads for 2 hours. **Fig 4A and 4B** show that treatment with LPS and NCEs significantly (p < 0.0001) increased the phagocytic activity of macrophage cultures, albeit the increase in phagocytic activity was ~ 2.5 fold greater for LPS than for NCEs. **Fig 4C** shows that treatment LPS and NCE increased also IL-1β secretion by macrophages. Also in this case the observed effect was stronger for LPS than NCEs.

Given that macrophages produce reactive oxygen species (ROS) during phagocytosis, we asked whether NCE stimulation also provoked increased ROS following NCE stimulation. As shown in **Fig 4D,** stimulation of macrophage cultures with NCE resulted in a significant increase in ROS, consistent with prior studies which have shown that TLR2 and TLR4 engagement are sufficient to by increasing mitochondrial ROS production [20].

## Discussion

The majority of tissue resident macrophages in the adult mammalian heart are established during embryonic development, and are maintained through self-renewal, rather than through blood monocyte input [14, 15]. However, after birth there is a partial dilution of the tissue resident embryonic-derived macrophages by monocyte-derived macrophages that arise in the bone marrow, and subsequently migrate to heart [14, 15]. Following tissue injury, there is a robust recruitment and expansion of monocyte derived macrophages to the heart [15, 21]. These monocyte-derived macrophages are then capable of undergoing classical (LPS) or alternative (IL-4) activation states, as they become established in areas of tissue injury in the heart.

One of the important questions with respect to the activation and polarization of the different macrophage phenotypes following myocardial injury is to better understand the specific molecular signals that are responsible for activating monocyte-derived macrophages as they enter the heart. Here we show that necrotic myocyte extracts contain proteins that skew the polarization of cultured macrophages towards a classical activation pattern characterized by upregulation of gene ontogenies that are associated with cell stress responses, as well as with inflammation (**Fig 1**). Intriguingly, the overlap in gene expression profiles for LPS stimulated and NCE simulated macrophage cultures was greatest within the first 24 hours after stimulation and decreased by 72 hours, at which time there was increased overlap of DEGs expressed

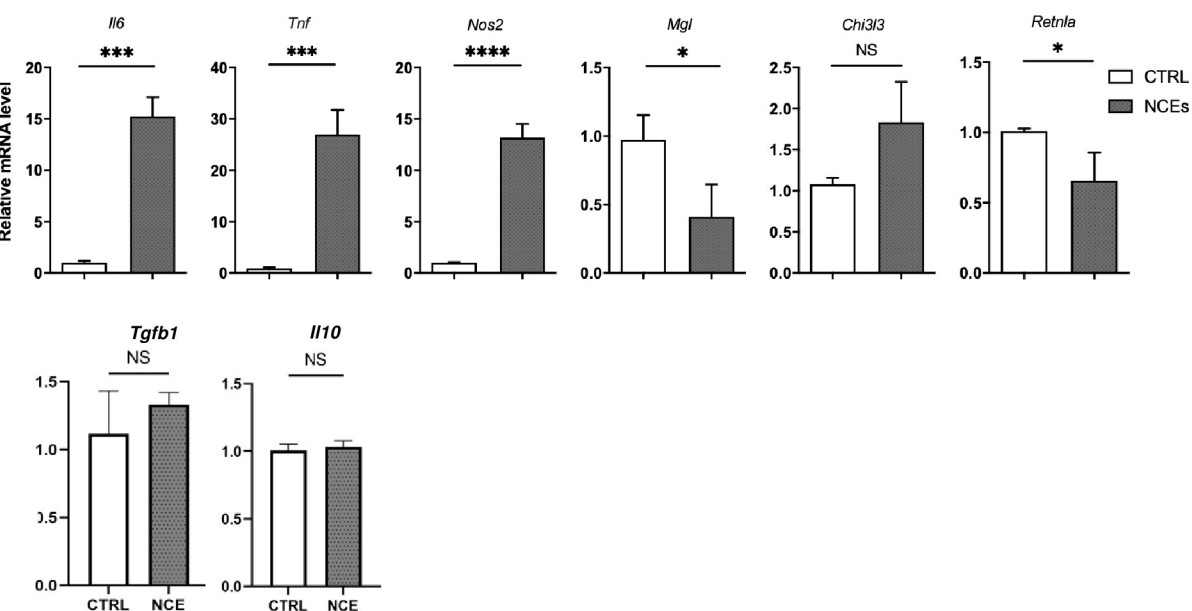

**Fig 2. RT-qPCR of macrophage genes.** RT-qPCR of changes mRNA levels of nitric oxide synthase 2 (*Nos2*), tumor necrosis factor (*Tnf*), interleukin-6 (*Il-6*), macrophage galactose-type lectin (*Mgl*), Chi3l3 (*Chi3l3*) and resistin like alpha (*Retnla*) in macrophage cultures that were stimulated with necrotic cell extracts (NCEs) or for 24 hours. RT-qPCR of changes mRNA levels for changes in transforming growth factor-β1 (*Tgfb1*) and interleukin-10 (*Il10*) were stimulated with necrotic myocardial extracts for 24 hours. All the mRNA expression levels were normalized to *36B4* mRNA levels and the fold-change in all genes were expressed relative diluent treated controls (n = 3 cultures/condition). Key: CTRL = control. LPS = , lipopolysaccharide. IL-4 = Interleukin-4. NCEs = necrotic cell extracts). (* = p<0.05 vs. control. ** = p<0.01 vs. control. *** = p<0.001 vs. control. **** = p<0.0001 vs. control; NS = non-significant vs. control).

in LPS, NCE and IL-4 stimulated macrophage cultures. These findings are consistent with the known plasticity of macrophages and suggest that NCE-induced skewing of macrophage polarization towards the classically activated phenotype is transient in nature. The functional significance of the NCE-induced macrophage skewing was demonstrated by the studies wherein NCE's provoked an increase in IL-1β secretion in cultured macrophages (**Fig 4C**),

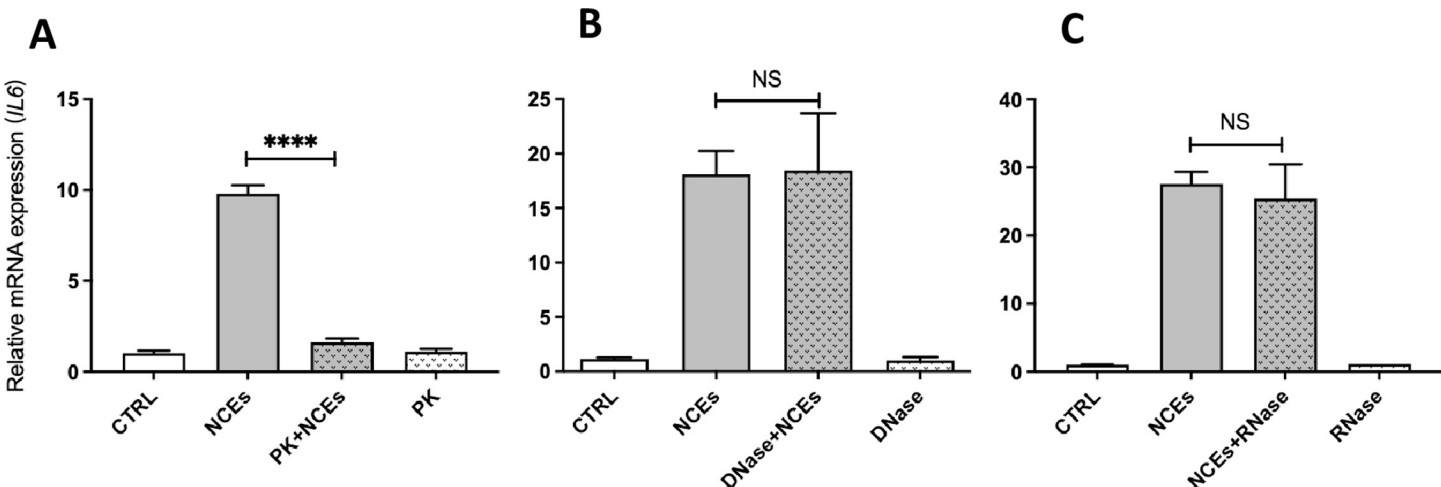

**Fig 3. Treatment of NCEs with proteinase K, DNAse and RNAse.** (A) Proteinase K digestion significantly reduces NCE-induced increase in *Il6* mRNA levels; (B, C) DNase or RNase pretreatment of NCEs had no effect on NCE-induced *Il6* mRNA levels (n = 3 cultures/condition). (key: CTRL = control. NCEs = necrotic cell extracts). (**** = p < 0.0001 vs. control; NS, non-significant vs. control).

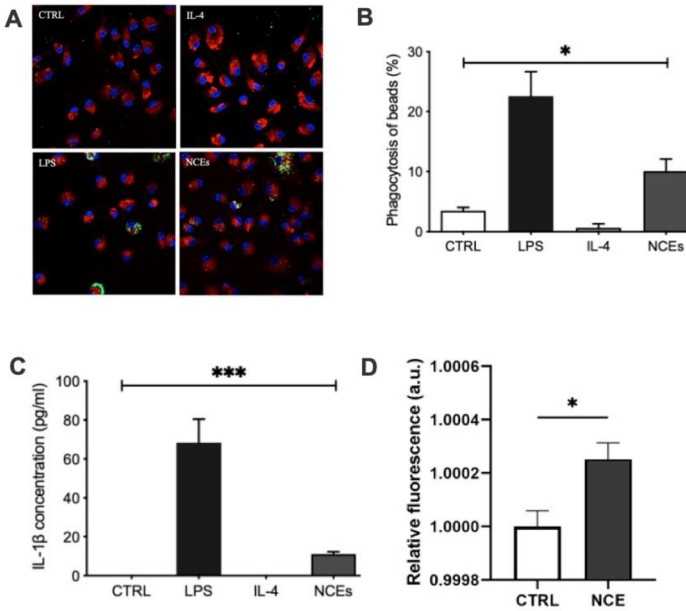

**Fig 4. Functional assessment of peritoneal macrophages stimulated with NCEs.** (A) Representative confocal images of uptake of fluorescent beads (green dots) by cultured CD64+ peritoneal macrophages (red) stimulated with diluent, NCEs, LPS or IL-4. DAPI (blue) was used to stain the nuclei (B) Group data for uptake of fluorescent beads by cultured peritoneal macrophages stimulated with diluent, NCEs, LPS or IL-4 (n = 3 cultures/treatment). (C) IL-1β levels in the supernatant of macrophage cultures stimulated with NCEs, LPS (n = 3 cultures/treatment). (D) Reactive oxygen species production in peritoneal macrophages. Primary peritoneal macrophages were stimulated with 10 μg/mL necrotic myocardial cell extract (NCE) for 24 hr or untreated (CTRL). A DCFDA assay was then performed to determine the amount of reactive oxygen species produced in NCE-treated cells relative to CTRL cells (n = 6). (key: CTRL = control (diluent); IL-1β = Interleukin-1β; IL-4 = Interleukin-4; LPS = lipopolysaccharide; NCEs = necrotic cell extracts). (* = p<0.05 vs. control. ** = p<0.01 vs. control. *** = p<0.001 vs. control. **** = p<0.0001 vs. control; NS = non-significant vs. control).

which is consistent with a classically activated macrophage state. In addition, we observed that NCEs were sufficient to stimulate increased phagocytosis and increased reactive species in the macrophage cultures (**Fig 4A and 4B**).

There are several limitations to the present study that warrant discussion. First, by design, we employed a simple in vitro culture system that would allow us to study the polarization of peritoneal-derived macrophage cultures in a controlled setting. It should be recognized, however, that the responsiveness and plasticity of thioglycolate stimulated peritoneal derived macrophage cultures may differ from that of monocyte-derive macrophages that enter the heart following tissue injury. Accordingly, further in vivo studies will be needed to confirm the relevance of our in vitro observations. Second, it should be recognized that the biological activity of DAMPs is exceedingly complex and depends on a variety of factors, including the overall extent of tissue injury, the type of cell death, and the type of cells dying. Here we used necrotic cell extracts from embryonic rat cardiac myocytes to stimulate cultured murine peritoneal derived macrophages. Given that the molecular motifs in necrotic cells that activate innate immune responses are highly phylogenetically conserved, the species differences between rat and mouse are unlikely to explain the observed findings in the present study. Nonetheless, the results of these studies should be regarded as provisional and will require validation using isogenic cell lines. Third, the scope of these studies was not designed to determine the specific proteins that were responsible for the observed NCE-induced macrophage polarization. This statement notwithstanding, we have shown previously that high mobility group box 1

(HMGB1), a non-histone nuclear protein that is known to skew monocyte derived macrophages towards an classically activate phenotype, are enriched in NCEs. Fourth, although we did not identify which receptors were responsible for NCE-induced macrophage skewing in the present study, we have shown recently that necrotic myocardial extracts activate macrophages through both TLR4 and TLR2 [22, 23]. Lastly, although the present study was focused on the role of necrosis on macrophage polarization, it should be recognized that efferocytosis of apoptotic cells by macrophages can lead to skewing towards an anti-inflammatory (i.e. M2) phenotype (reviewed in [24]).

## Conclusions

Using a very simple well-defined in vitro culture system we show that proteins released from necrotic myocardial cells acutely promote macrophage polarization towards an activated pro-inflammatory phenotype. These findings suggest that necrotic material released from dying cardiac myocytes in the context of acute myocardial infarction may contribute to the polarization of monocyte-derived macrophages that are recruited to the sites of injury. Further work will be needed to verify these findings in vivo and to assess the potential therapeutic importance of modulating NCE induced macrophage polarization in patients with ischemic myocardial injury.

## Supporting information

**S1 File. List of differentially expressed genes that were up and down regulated following stimulation with NCEs or LPS.**
(XLSX)

## Author Contributions

**Conceptualization:** Wenlong Jiang, Luigi Adamo, Sarah Evans, Cibele Rocha-Resende.

**Data curation:** Scot J Matkovich.

**Methodology:** Wenlong Jiang, Kenji Lim, Scot J Matkovich, Sarah Evans, Cibele Rocha-Resende, Douglas L Mann.

**Project administration:** Wenlong Jiang, Kenji Lim.

**Writing – original draft:** Wenlong Jiang.

**Writing – review & editing:** Luigi Adamo, Douglas L Mann.

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
