## [Decision Letter · Decision Letter 0]

9 Nov 2022

PONE-D-22-27365Necrotic cardiac myocytes skew macrophage polarization towards a classically activated phenotypePLOS ONE

Dear Dr. Jiang,

Thank you for submitting your manuscript to PLOS ONE. After careful consideration, we feel that it has merit but does not fully meet PLOS ONE’s publication criteria as it currently stands. Therefore, we invite you to submit a revised version of the manuscript that addresses the points raised during the review process.

Your manuscript was reviewed by two experts and we received positive feedback. I would like to suggest authors to demonstrate the properties of proteolytic lysate by western blot using cardiomyocytes markers as well as standard Coomassi Blue  or similar staining of the same gel. I am curious whether varying amount of PK or other factors has any  association with macrophage polarization. 

We look forward to receiving your revised manuscript.

Kind regards,

Partha Mukhopadhyay, Ph.D.

Section Editor

PLOS ONE

Journal Requirements:

Reviewers' comments:

Reviewer's Responses to Questions

**Comments to the Author**

1. Is the manuscript technically sound, and do the data support the conclusions?

Reviewer #1: Partly

Reviewer #2: Yes

2. Has the statistical analysis been performed appropriately and rigorously? 

Reviewer #1: Yes

Reviewer #2: Yes

3. Have the authors made all data underlying the findings in their manuscript fully available?

Reviewer #1: Yes

Reviewer #2: No

4. Is the manuscript presented in an intelligible fashion and written in standard English?

Reviewer #1: Yes

Reviewer #2: Yes

5. Review Comments to the Author

Reviewer #1: In the present study “Necrotic cardiac myocytes skew macrophage polarization towards a classically activated phenotype” the authors discussed the macrophage polarization pattern after the stimulation of necrotic cardiac myocyte extraction body. Here are some major concerns to be addressed:

1. The current study did not distinguish the NCE from cardiomyocytes from other cells, so it is unclear whether the current effect is specific to myocardial infarction of universal. The authors should make NCEs from other cells and compare the effects.

2. The current study did not analyze the receptor to mediate the current effect. Different PRR receptor inhibitors should be used to analyze which PRRs are involved.

3. The authors found that protein component is responsible for the classical phenotype polarization. However, more detailed information should be obtained. At least the protein from nucleus, mitochondria and cytosol should be separated to be analyzed in the current polarization study.

4. Reactive oxygen special production should be studied.

5. The alternative polarization markers are not typical. The authors should use better recognized ones such as IL-10, CD206, Arginase and TGF-beta

6. The authors should prepare apoptotic bodies to compare the polarization pattern, which also proves the capacity of the macrophages for alternative polarization.

Reviewer #2: In this manuscript, Jiang et al., showed that proteins released from NECs skew the polarization of macrophages to classically activated phenotype. For the benefit of the readers, here are my comments for further improvement of this manuscript.

1) I request the authors to mention that in line 190, the number of differently expressed genes for "LPS only" is 1878. Because from the Venn diagram the total number of differently expressed genes for LPS is 1878+77+123+1205 which is 3283. I request the authors to update the text accordingly in lines 190-192 and 195-196.

2) I request the authors to write the magnitude of each bar in figure 1B because with the current y-axis scale for NCE and IL-4 panel, and NCE and LPS+IL-4 panel, it is very difficult to know the height of these bars.

3) In figure 1C, can the authors provide some example genes for each of the ten conditions?

4) It will be helpful if the authors can comment on why they did not see any changes in the Chi3l3 mRNA levels in figure 2.

5) It is not clear how the information presented in figure 3 is novel. I request the authors to mention some examples from literature where DNA, RNA, and proteins are shown to be involved in macrophage polarization.

6) I request the authors to label the panels in figure 4. Currently, it does not say which panel is A, B, and C.

7) In line 250 of the discussion section, the authors mention that necrotic myocyte extracts contain proteins. Can the authors make some predictions on what proteins might be playing a role in this mechanism?

8) In the figure legends why did the authors say S1 Fig 1? This is not a supplementary figure in the manuscript. Hence, I request the authors to just say Fig. 1. I also request the authors to make the same correction for the legends of Fig. 2-4.

9) In figure 4, I request the authors to mention what each color represents in the figure caption.

10) There is a typo in line 63 of page 3. I think the authors meant completely instead of complexly.

6. PLOS authors have the option to publish the peer review history of their article (what does this mean?). If published, this will include your full peer review and any attached files.

Reviewer #1: No

Reviewer #2: No

---

## [Author Response · Author response to Decision Letter 0]

16 Jan 2023

Thanks you for all the comments. Since the answers are too long, I created a document to answer all the comments calls "DAMP_Macs rebuttal“. Appreciate!

---

## [Decision Letter · Decision Letter 1]

27 Feb 2023

Necrotic Cardiac Myocytes Skew Macrophage Polarization Towards a Classically Activated Phenotype

PONE-D-22-27365R1

Dear Dr. Jiang,

We’re pleased to inform you that your manuscript has been judged scientifically suitable for publication and will be formally accepted for publication once it meets all outstanding technical requirements.

Kind regards,

Partha Mukhopadhyay, Ph.D.

Section Editor

PLOS ONE

Additional Editor Comments (optional):

Reviewers' comments:

Reviewer's Responses to Questions

**Comments to the Author**

1. If the authors have adequately addressed your comments raised in a previous round of review and you feel that this manuscript is now acceptable for publication, you may indicate that here to bypass the “Comments to the Author” section, enter your conflict of interest statement in the “Confidential to Editor” section, and submit your "Accept" recommendation.

Reviewer #1: All comments have been addressed

Reviewer #2: All comments have been addressed

2. Is the manuscript technically sound, and do the data support the conclusions?

Reviewer #1: (No Response)

Reviewer #2: (No Response)

3. Has the statistical analysis been performed appropriately and rigorously? 

Reviewer #1: (No Response)

Reviewer #2: (No Response)

4. Have the authors made all data underlying the findings in their manuscript fully available?

Reviewer #1: (No Response)

Reviewer #2: (No Response)

5. Is the manuscript presented in an intelligible fashion and written in standard English?

Reviewer #1: (No Response)

Reviewer #2: (No Response)

6. Review Comments to the Author

Reviewer #1: (No Response)

Reviewer #2: (No Response)

7. PLOS authors have the option to publish the peer review history of their article (what does this mean?). If published, this will include your full peer review and any attached files.

Reviewer #1: No

Reviewer #2: No

---

## [Editor Report · Acceptance letter]

23 Mar 2023

PONE-D-22-27365R1 

Necrotic Cardiac Myocytes Skew Macrophage Polarization Towards a Classically Activated Phenotype 

Dear Dr. Jiang:

I'm pleased to inform you that your manuscript has been deemed suitable for publication in PLOS ONE. Congratulations! Your manuscript is now with our production department. 

Kind regards, 

on behalf of

Dr. Partha Mukhopadhyay 

Section Editor

PLOS ONE